# From Plants to Wound Dressing and Transdermal Delivery of Bioactive Compounds

**DOI:** 10.3390/plants12142661

**Published:** 2023-07-16

**Authors:** Gabriela Olimpia Isopencu, Cristina-Ileana Covaliu-Mierlă, Iuliana-Mihaela Deleanu

**Affiliations:** 1Department of Chemical and Biochemical Engineering, University Politehnica of Bucharest, Polizu Str. 1-7, 011061 Bucharest, Romania; g_isopencu@chim.upb.ro; 2Department of Biotechnical Systems, Faculty of Biotechnical Systems Engineering, University Politehnica of Bucharest, 313 Splaiul Independentei, 060042 Bucharest, Romania; cristina_covaliu@yahoo.com

**Keywords:** bioactive molecule, biomaterials, transdermal delivery, cell wall components, metabolites, antimicrobial, wound dressing

## Abstract

Transdermal delivery devices and wound dressing materials are constantly improved and upgraded with the aim of enhancing their beneficial effects, biocompatibility, biodegradability, and cost effectiveness. Therefore, researchers in the field have shown an increasing interest in using natural compounds as constituents for such systems. Plants, as an important source of so-called “natural products” with an enormous variety and structural diversity that still exceeds the capacity of present-day sciences to define or even discover them, have been part of medicine since ancient times. However, their benefits are just at the beginning of being fully exploited in modern dermal and transdermal delivery systems. Thus, plant-based primary compounds, with or without biological activity, contained in gums and mucilages, traditionally used as gelling and texturing agents in the food industry, are now being explored as valuable and cost-effective natural components in the biomedical field. Their biodegradability, biocompatibility, and non-toxicity compensate for local availability and compositional variations. Also, secondary metabolites, classified based on their chemical structure, are being intensively investigated for their wide pharmacological and toxicological effects. Their impact on medicine is highlighted in detail through the most recent reported studies. Innovative isolation and purification techniques, new drug delivery devices and systems, and advanced evaluation procedures are presented.

## 1. Introduction

The skin, as the body’s largest and most exposed organ, can provide a readily accessible delivery route for therapeutic substances. In fact, the topical application of remedies has been practiced for thousands of years. From Galen, the Father of Pharmacy, who introduced herbal ingredients into the formulation of drugs, the evolution of skin wound dressing/healing (WD) and transdermal delivery (TD) took an interesting path, which continues, and grows nowadays with the help of modern scientific techniques [1].

Most topical drug delivery systems (DDSs) were traditionally designed for the local administration of active pharmaceutical ingredients (APIs), with numerous applications in the field of skin tissue healing [2,3,4]. However, the concept of soft (permeable) drugs that can also affect the tissues/organs beneath the skin originates from Ibn Sina (Avicenna), the great Persian physician and philosopher, born in 980 AD in the village of Afshanah [5,6]. This is now believed to be the oldest conceptualization of transdermal drug delivery (TDD).

Currently, there is a constant search for new materials and new medical devices (wound patches, scaffolds, and transdermal delivery systems) that can provide effective treatment through the unique, controlled, and targeted delivery of APIs. New methods are sought to replace conventional drug delivery routes (gastric, intravenous, intradermal, etc.) with TDs due to their numerous advantages: allowing the direct access of APIs to the bloodstream through the skin and avoiding liver metabolism and the gastrointestinal tract; allowing the homogenous absorption of APIs and increasing pharmaceutical effectiveness; reducing/eliminating drug alterations that can occur in the interaction with patient enzymatic and immune systems; and, equally important, TD is more comfortable, non- or less invasive, and can be easily self-administered [3,7,8,9]. Furthermore, through bionanotechnology, sophisticated devices can be designed to allow drug storage and controlled, on-demand delivery for several days [6,9].

However, of course, there are some disadvantages as well, and major drawbacks that limit, in many cases, the utilization of natural compounds in topical and TDD systems related to both skin and natural compound properties.

As will be briefly detailed further, development strategies involve two major directions: (i) increase skin permeation using chemical penetration enhancers or other more or less invasive methods (electrical, thermal, mechanical, and others), or a combination of methods (microneedles combined with iontophoresis) [10], and (ii) an increase in the bioavailability, stability, and biodistribution of these natural compounds [11].

This review work summarizes the newest outcomes that involve natural compounds of a plant origin as constituents of novel dermal and transdermal drug delivery systems. Major benefits and also important drawbacks associated with their extraction and isolation, as well as stability and bioavailability, are emphasized. This review presents new technologies and modern techniques developed to minimize traditionally known barriers like the complexity of applications, potential adverse effects, and proof of functional changes.

Synthetically, plant constituents, with or without biological activity, are classified based on the metabolic pathways that produce them, and their role in developing TDDS is presented.

## 2. Natural Compound Transfer Specifics and API Delivery Methods

Dermal and TDD methods, disregarding the principles and formulations, are directly related to skin and drug/API properties, as already mentioned.

### 2.1. Skin Properties and Drug Transport Mechanism

The skin is a complex three-layer structure comprised of the epidermis, dermis, and hypodermis. It is a two-way protective barrier preventing any external intrusions (microbial, chemical) and, at the same time, the excessive loss of endogenous material. Skin permeability, determined not only by its structure but also by numerous factors like the anatomic site, gender, age, hydration, or external temperature, is mainly associated with *Stratum Corneum* (SC, the outermost layer), containing a high amount of keratin [7].

Simplistically, TDD means API penetration through the skin at therapeutic rates, but overall, the process, also called percutaneous absorption, is more complicated, described in five main stages. A “vehicle” (transdermal device, cream, or solvent) is often used to incorporate the drug/API. In the first stage, the active molecule is distributed from the vehicle to the hydrolipidic film based on the partition coefficient and molecule’s mobility [12]. The drug is then absorbed into a specific skin layer (penetration step) and partitioned from SC into the aqueous viable epidermis, from where the molecule diffuses to the upper dermis. The permeation from one layer to another further takes place, and finally, the drug enters the bloodstream (the resorption or absorption step) [13,14,15]. Each stage depends on numerous factors and can be differently kinetically characterized, but overall, the transfer is controlled by the molecule’s passive diffusion [12].

In the case of skin wound healing, the mechanisms are more or less the same, as the API is also entering into the bloodstream, but the required pharmaceutical effect is local.

Two major pathways are known to allow some molecule transfer: transepidermal, when a drug permeates into the cells (intracellular) or through the cellular interspaces (transcellular), and transappendageal (via hair follicles or sweat glands) [15,16]. Disregarding the mechanism or the pathway, there are certain physicochemical properties that a potential drug/bioactive substance must meet for a TD, such as [6,17]:-Non-toxicity.-Low-dose administration requirements.-High lipophilicity.-Molecular weight below 500 Da.-Hydrophilic/hydrophobic balance corresponding to an n-octanol/water partition coefficient (log P) between 1 and 5 (which corresponds to moderate lipophilicity).-Acceptable water solubility (0.05 to 1 mg/mL).-Melting point below 250 °C.

These requirements, which would naturally allow successful TDD, are also known as the main factors associated with the limited usage and commercial applications of bioactive TDDS up until now.

### 2.2. Conventional and Modern Topical and TDD Methods

Depending on the technique(s) that allows API to permeate and/or cross the skin, TDD methods are classified as passive or active.

Passive methods, also known as conventional or chemical methods, which include “passive” patch technology, are based on the passive permeation of bioactive molecules. Often these methods are improved via the implementation of penetration/permeation/skin enhancers (usually a chemical or biochemical substance) or supersaturated systems, or other ways, to enhance the mass transfer driving force or to improve skin permeability, allowing for improving dose control. However, TDDs are not fundamentally changed, and the applicability of passive systems remains limited [12,18]. Furthermore, additional drawbacks are associated with permeation enhancer usage [17]:-Toxicity and skin irritation.-Two-way skin permeabilization (increase in transepidermal water loss).-Additional pharmacological effects.-Specificity towards a certain drug.

Active (physical) methods, as accurately described by Alkilani et al., “involve the use of external energy to act as a driving force for drug transport across the skin or by physically disrupting the SC” [15]. There is a need for an enhanced TDD, derived naturally together with biotechnological and bioengineering progress, when new (bio)active substances, macromolecules in general, characterized by a low resistance in the gastrointestinal tract, become available for medical and cosmetic therapies [18]. Since the introduction of the first-generation TDDs at the commercial scale in the 1950s, significant advances have been made, and numerous active methods have been tested and/or implemented, as depicted in Figure 1 [16,19]. Among these, the most applied are electrical techniques like electroporation and iontophoresis, mechanical systems like microneedles, or ultrasound-assisted TDDs. Of course, these techniques are not without disadvantages.

High costs, large-scale feasibility, limited loading capacity, stability, or other issues in controlling drug delivery are determining the search for even more advanced techniques. Most recent studies involve integrated chemical and physical approaches (hybrid methods) [20,21].

## 3. Bioactive Transdermal Patches and Wound Dressing Materials

Transdermal patches (TP) and WD materials are widely used nowadays. These are designed in many forms and configurations. From ancient formulations consisting in ointments bandaged on the skin (Figure 2a1) to state-of-the-art microneedle patches (Figure 2c), the evolution can be considered spectacular [22,23]. One of the first described TDDs (schematically represented in Figure 2a3), and one of the first studies to prove that the limiting step of drug transfer is the diffusion through the skin layer, was introduced by Wurster and Kramer early in 1960, 10 years before the first patent involving continuous drug-rate-controlling delivery was filled [22,24].

By definition, bioactive means having a certain effect on living organisms [25]. A bioactive transdermal patch (BTP) or a WD material can be a structure of a certain complexity to allow the delivery of a bioactive molecule through the skin, directly to the bloodstream. Disregarding the structural complexity, such a patch consists of the following [22,26]:The active component/drug, which could be a natural or a synthetic component with bioactive properties; it can be supplied from a reservoir or from another structure of the patch [27].Backing material, usually made of elastomers, is used to provide patches’ flexibility and protection from the outer environment and water.Drug-releasing membrane/matrix, made of natural/synthetic polymers/elastomers.Adhesive, to keep the patches’ layers together and adhered to the skin; it may also contain the permeation enhancers and/or the drug.The protective liner.

Countless materials/substances are used as either one of these devices’ structural elements, and natural products of a plant origin are included. In fact, medicinal plants have always been an important part of the global medicinal sector. Traditionally, these were used for their curative properties, but nowadays, there are many other ways to use plant materials in the medical field (in general), and as a TP component (in particular). Many examples can be given regarding the use of cell walls’ structural components or plants’ metabolites as constituents in TP development. In our review, we will consider only the most recent ones (with some exceptions), since other reviews cover older investigations.

## 4. Natural Products from Plants Used in Medical Devices

In a wide meaning, the natural product (NP) is a chemical compound produced by life (including biotic materials, bio-based materials, bodily fluids), while more restrictively, a natural product is an organic compound produced by a living organism [28,29]. Most NPs are highly complex structures, with a large number of sp^3^ carbon and oxygen atoms and low-nitrogen and halogen atoms, low cLogP values, and molecular rigidity [30]. These characteristics can be regarded, depending on application specificity, as beneficial in the search for new NP-based drugs and formulations.

Simplistically, NPs derived from plants are classified, based on the metabolic pathways that produce them, as primary or secondary metabolites, although there is no clear boundary between them. Their specific metabolic role, structure, and properties allow them to be used as constituents in TP, as will be further detailed.

### 4.1. Primary Metabolites as Constituents of BTP

Primary metabolites (PM) are organic compounds responsible for photosynthesis, respiration, and plant development (with intrinsic functions). PM include core molecules like amino acids, organic acids, and unsaturated fatty acids, and major macromolecules built based on these: carbohydrates, lipids, and proteins [31]. PM are used in transdermal formulations mainly as constituents of the releasing matrix, and only in a few cases as bioactive components.

In recent years, a great interest has been given to plant mucilages and gums as excellent sources of plant-based primary compounds.

#### 4.1.1. Plant Mucilages

A plant mucilage is a hydrocolloid, containing mainly carbohydrates/monosaccharides, amino acids, glycoproteins, and uranic acid units, in varying concentrations and compositions, depending on the plant source. For a plant, the mucilage represents a vital constituent of the cell, or a part of the cell walls, and can be found in almost all its parts, including the seeds [32,33]. For humans, a mucilage was traditionally used as a gelling and texturing agent in the food industry. In time, it proved to be a valuable and cost-effective natural component source in the biomedical field as well [34]. It is now used as a biodegradable, biocompatible, and non-toxic rate-controlling and matrix-forming agent in TDDS and WD applications, as will be briefly illustrated in the following examples.

In the early stages of DDS, plant mucilages were used as mucoadhesive polymers, due to their chemical characteristics (possessing numerous carboxyl and hydroxyl groups). Conducted studies involving a *Mimosa pudica* (family: Mimosaceae) mucilage showed positive effects on both the bioadhesion time and drug release percentage [35]. Similar beneficial observations were recently reported for *Datura stramonium* leaves’ mucilage. Ahad et al. found in a factorial organized study that the mucilage concentration influences the rate and the quantity of Aceclofenac release and also the adhesion time [36].

An interesting valorization of a plant mucilage’s properties (especially charge density here) involves electrospinning, a modern technique applied to obtain nanofibers. Nanofibrillated mats and patches are considered ideal wound healing materials, allowing a sufficient oxygen supply and facile incorporation of the API [37]. A *Plantago ovata* mucilage was used as a precursor for the preparation of core–shell nanofibers. Biodegradable stimuli-responsive core–shell nanofibers were prepared using the mucilage as a shell solution (outer needle) and PTX-loaded PCPP-CA micelles (Paclitaxel-loaded Cholic acid conjugated to poly (bis (carboxyphenoxy)phosphazene)) as a core solution (inner needle) [38]. This complex nanomaterial, assembled as a patch, was developed to achieve the transdermal controlled delivery of PTX in breast cancer therapy. In vitro studies revealed 180 h of pH-responsive and sustained drug release, and thus, an improved anticancer efficacy compared to free PTX.

In a similar approach, a unique combination of *Hibiscus rosa-sinensis* leaves’ mucilage (HLM)–PVA–Pectin was electrospun and crosslinked with glutaraldehyde vapor [37]. The obtained “green mat” can be used as a nanofibrillated polymeric scaffold for wound healing. This is just an example of how natural and synthetic polymers can be combined to obtain a hemocompatible and biodegradable product.

Although the use of plants’ mucilage was not that often reported between 2010 and 2020, and their applicability was rather limited, the shift to “natural” sparked a research influx recently. Furthermore, modern techniques of extraction/isolation, characterization, and investigation allowed the identification of secondary metabolites in mucilages, especially when obtained from foliar parts of a plant [39]. Thus, in some cases, a plant mucilage is used solely as a biopolymer with pharmacological activities (antimicrobial, antioxidant), and in other cases, it is the source of the API in a more complex matrix, as can be seen in Table 1.

#### 4.1.2. Plant Gums

Gums are polysaccharides of multiple sugars, with numerous applications in food, pharmaceutical, and medical applications. In most cases, they are naturally occurring, as a result of plants’ protection mechanisms [58]. These are solid in dry environments, and gluey in the presence of humidity. In fact, a major drawback of gums’ usage derives from their uncontrollable rate of hydration and readily-water-soluble characteristic.

Plant-derived gums can be used directly or chemically modified, aiming for customized characteristics of the product. In reported applications, plant gums were used as excipients or matrix-forming materials in mucoadhesive polymeric systems or medical tablets [59,60]. Results indicate that gums could improve mucoadhesive strength or could be a key factor in controlling drug delivery. In a general approach, aiming for applicability not only in pharmaceutical but also in food, medical, or cosmetic industries, versatile hydrogels and biopolymeric nanocarriers were developed based on tragacanth gum or moringa bark gum, as promising controlled DDS [61,62,63,64,65].

Guar gum, which can be obtained from the seeds of *Cyamopsis tetragonoloba*, was grafted and used in TDD. Thus, membranes for the transdermal controlled release of diclofenac were fabricated using guar gum grafted with acrylic acid, and nanosilica. Slow and sustained drug delivery was achieved due to cage morphology and hydrophobicity in the case of nanocomposites with 1 wt% nanosilica content [66]. In another study, poly(N-isopropylacrylamide)-grafted guar gum, incorporating cellulose nanofibers (CNF), was used for the TD of diltiazem hydrochloride. The obtained biomaterial proved to be a non-irritant to the skin and non-toxic. Similar to nanosilica, 1 wt% CNF determined optimal thermal, mechanical, and barrier properties. A sustained drug release was obtained for several hours (approx. 23% release after 20 h) [67].

In a different approach, cashew gum extracted from *Anacardium occidentale* was acetylated, and the obtained solution was used to prepare drug-loaded nanoparticles with nanoprecipitation and dialysis techniques [68]. Nanostructures containing diclofenac diethylamine were designed as drug carriers for TD. Both applied methods allowed an efficient drug encapsulation and in vitro 90% TD permeation was obtained in 6 h. Moreover, the TDD could be controlled.

The complex composition of these plant-derived biopolymers allows them, as emphasized by the described applications, to be successfully used in numerous fields as emulsifying, gelling, stabilizing, binding, and filming agents, or even as active functional ingredients [69]. As promising as it may be, the usage of mucilages and gums has some important limitations to be considered and overcome, aiming for industrial applicability. Disregarding the plant source/plant parts, their composition, which is always very complex, may differ due to climate conditions, regional characteristics, agricultural and agronomical variations, local availability (in some cases), and preparation techniques. The storage conditions must be very carefully addressed, to minimize or eliminate the risk of contamination and microbial spoilage, thermal degradation, and natural biodegradability.

#### 4.1.3. Plant Cell Wall Components

A plant cell wall can be defined as the greatest renewable resource available. It is a complex, heterogenous structure, comprised of more than 90% polysaccharides, specifically assembled, depending on plant species and tissue types [70,71]. Major cell wall structural components that are mainly used as excipients in medical devices are cellulose, hemicellulose, and pectin.

***Cellulose***, the main component of cell walls of any plant, is chemically defined as unbranched β-(1,4)-linked glucan chains. It presents in a complex inhomogeneous assembly due to multiscale linear fibrils formed during numerous and various biosynthesis processes [71]. As a water insoluble carbohydrate, it was extensively used in many industrial fields. In modern pharmaceuticals, cellulose (as fibrils, microcrystals, or nanocrystals) and cellulose derivates obtained with chemical routes are often used as excipients for the development of controlled DDS [3,72,73,74].

Cellulose nanofibers (CNf), obtained with mechanical methods or pretreatments, are used to increase matrix hydrophilicity, biocompatibility, stability, and mechanical strength [75,76]. Moreover, CNf high porosity allows the facile preparation of 3D network hydrogels that can embody drugs or other active nanoparticles [77]. Similarly, to increase mechanical properties of WD materials, microcrystalline cellulose (MCC) can be used. For instance, Ding et al. prepared a novel fiber-type dressing material composed of alginate, polyvinyl alcohol, MCC, and *Euphorbia humifusa* Willd extract by microfluidic spinning. The obtained medical dressing proved to have good mechanical properties due to MCC and an antibacterial effect due to the plant extract [78].

Carboxymethyl cellulose (CMC) is the first and maybe the most used among cellulose derivates, due to its characteristics (cost effectiveness, biocompatibility, stability, stimuli responsiveness, and water solubility). Numerous applications and materials were developed based on CMC [79]. Some of the most recent can be named (2023 publication year):-Self-healing biodegradable 3D network hydrogel, based on boronic-acid-grafted CMC, used as a drug release vehicle or scaffold for tissue regeneration or bleeding control [80].-Antibacterial photo-inspired WD, based on CMC as the “network backbone” hydrogel, polyvinyl alcohol (PVA) with iodophor (KI-I_2_) as the antimicrobial hydrogel, and a fluorescent material (carbon quantum dots) as the doping agent. This functional material releases desired doses of an antibacterial agent (I_2_), and provides real-time information on the wound healing process, based on hydrogel color responsiveness to wound pH changes in UV and visible light [81].-Multilayer microneedles, based on PVA and CMC for the transdermal delivery of nicotinamide mononucleotide (antiaging agent). CMC properties, as a high water solubility and hydrophilicity, determine a higher drug release [82].-Dissolvable patch, based on sodium alginate and CMC, loaded with cephalexin monohydrate as an antimicrobial agent for skin and chronic wound infections. A sustained drug release is obtained for selected concentrations [83].-Chromogenic aerogel-like composite, based on CMC and PVA with an anthocyanin extract immobilized into the polymeric matrix, as a color indicator for wound pH and antimicrobial agent at the same time [84].

Other WD materials were developed based on cellulose acetate [85,86,87,88], hydroxyethyl cellulose [89], ethyl cellulose [90], hydroxypropyl methylcellulose [91], or cellulose micro/nanocrystals [78,92] (references are of this year). An interesting approach in developing vegetable patches was reported by Buonvino et al., who used the *Lupinus albus* L. hull, a vegetal waste, as raw material for the development of a novel active bioplastic, to be used as WD material or a cellular scaffold, benefiting from both cellulosic and phytochemical content of the hulls [93].

***Hemicellulosic polysaccharides*** (β-glycan backbones) are used in developing WD, due to their biocompatibility and biodegradability, but not as much as cellulose [94]. Recently, Li et al. used hemicellulose as a reinforcing component in developing a flexible hydrogel with exceptional characteristics, to be used as self-powered TDD with motion sensing capacity [95].

***Pectins*** are cell wall polysaccharides with α-(1,4)-linked galacturonic acids in their backbone [71]. Their distinctive properties among biopolymers, such as gelling and mucoadhesivity, combined with the resistance to amylases and proteases, encouraged pectin-based nanomaterial development for numerous medical applications, including WD [96]. Pectins are used in a native form or chemically modified. Thus, hydrogels composed of quaternized chitosan and pectin from a citrus (family Rutaceae) peel, containing propolis as an antimicrobial factor, are proposed as WD materials. Both pectin and propolis concentration influenced propolis release profiles, film mechanical properties, and stability [97]. Based on the expertise regarding extensive research, Raghavendra Naveen et al. obtained the extended release of neomycin sulphate from a novel carrier transdermal system—a polymer-layered silicate nanocomposite, developed using polyethyleneglycol, montmorillonite, and thiolated pectin. For an optimized composition, almost 90% of neomycin is released in 18 h. The nanocomposite exhibits the desired properties for a successful TDD (water vapor permeability, swelling, non-cytotoxicity) [98].

### 4.2. Secondary Metabolites as Constituents of BTP

Based on PM, plants secrete some biologically active chemicals, which are not involved in plants’ development and growth, called secondary metabolites (SM). The function of these SM is mainly to prevent diseases in living organisms, although the function of many of them still remains unknown. Secondary plant metabolites can be classified into four major classes [31,99]: **alkaloids, phenolic compounds, sulfur-containing compounds**, **and terpenoids**.

In general, TP containing plant extracts are composed of active principles that activate blood circulation, together with strong irritant substances. It is thus essential to control the concentration of these compounds in the TD system, so that the optimal clinical dose is delivered.

#### 4.2.1. Alkaloids

***Alkaloids*** are a group of nitrogenous heterocyclic bases of a low molecular weight, obtained from different parts of a plant. They are residual alkaline products of plant metabolism, mainly produced in defense mechanisms. Alkaloids present a large spectrum of pharmacological effects and are used as medications or as recreational drugs. Depending on the pharmaceutical effects, they can be classified as local anesthetics and stimulants (cocaine), psychedelics (psilocin), neuro-stimulants (caffeine, nicotine), analgesics (morphine), antibacterial (berberine, kokusaginine, nkolbisine), anticancer drugs (vinblastine, vincristine), antihypertensive agents (reserpine), cholinomimerics (galantamine), spasmolysis agents (atropine), vasodilators (vincamine), antiarrhythmia (quinidine), antiasthma therapeutics (ephedrine), and antimalarials (quinine) [100].

However, not all of these alkaloids can be used as API in TD. Those that are already integrated into medical instruments (TP), or those studied to be used for transdermal treatments, are listed as follows.

***Scopolamine*** is a tropane alkaloid produced by the Solanaceae (nightshade) plant family, such as henbane (*Hyoscyamus niger*), devil’s trumpet (*Datura stramonium*), angel’s trumpet (*Brugmansia versicolor*), deadly nightshade (*Atropa belladonna*), mandrake (*Mandragora officinarum*), and corkwood (*Duboisia myoporoides*) [100]. It was the first approved active substance impregnated in a TP for commercial use (by FDA in 1979). It is used to ameliorate motion sickness and nausea [101,102] or for its anticholinergic action [103]. The principle of passive transport through skin layers was maintained unchanged until the present, as well as the following manufacturing principles [104]:(a)An adhesive layer that sticks to the skin and contains scopolamine in a polymeric gel, which provides an initial priming dose.(b)An intermediate microporous polypropylene rate-controlling membrane.(c)The scopolamine reservoir that sustains a zero-order input of a drug to the skin surface.(d)A backing of impermeable aluminized polyester film.

***Opiates (morphine, codeine)*** are isoquinoline alkaloids extracted or refined from natural plant matter—poppy sap and fibers (*Papaver somniferum*). Since morphine is among the first analgesic substances used, it is also the first tested for TD, to reduce the metabolic effects of ingestion, or the discomfort of intravenous injection. Westerling, 1994 proposed a rudimentary transdermal morphine delivery system. The system involves bypassing the SC by actually removing the SC using a vacuum. This leads to the formation of a lesion on the skin, over which a diffusion chamber containing gauze impregnated with morphine is applied. The diffusion yield is 75% in the first 11 h after application [105]. Although the diffusion rate is quite low and the clinical effect is also low, TP with morphine have been developed. Thus, a TP made of polyethylene sponge foam as the retaining agent was designed to be applied directly to the skin. Morphine hydrochloride is delivered without any invasive permeation enhancer or a rate-limiting membrane. An adhesive layer to attach the patch to the clean, non-hairy dermal surface is also involved [106,107].

***Caffein*** is a purin-alkaloid found in more than 60 species of plants across the globe. Caffeine comes from the seeds of coffee beans (*Coffea arabica* and *Coffea robusta*), cacao beans (*Theobroma cacao*), and Kola nuts (*Cola acuminata*, *Cola nitida*); the leaves and buds of tea; the leaves of Yerba mate (*Ilex paraguariensis*); or the bark of Yoco (*Paullinia yoco*) [100]. This alkaloid acts as a vasodilator involved in numerous nervous system disorders: migraine, epilepsy, traumatic brain injury, ischemia, anxiety, alcoholism, Alzheimer’s disease, and brain cancer [108,109]. Because of a low solubility in the lipid layer of the SC, caffeine can be a TD using different penetration enhancers [110,111], or physical penetration methods to cross the SC of the skin (mechanical, ultrasounds, ionophoresis, microneedles) [108,111,112,113,114,115].

***Nicotine*** is a potent parasympathomimetic alkaloid mostly found in Solanaceae. It is found in a high concentration (approximately 0.6 to 3.0% of the dry weight) in *Nicotiana spp*. (tobacco), and is present in the range of 2 to 7 μg/kg in various edible plants of the Solanaceae family such as tomatoes (genera *Lycopersicon*), potatoes and eggplants (genera *Solanum*), and peppers (genera *Capsicum*) [100]. There are a multitude of commercial TP containing nicotine, used in particular to reduce smoking addiction or migraines. The active substance is incorporated in the form of a drug reservoir, or impregnated in a matrix [26,116].

***Berberine hydrochloride*** (***BH***), or berberine, is an isoquinoline alkaloid extracted from several plants of the genus Berberis such as *Berberis aquifolium*, *Berberis vulgaris*, *Berberis aristata* (*Berberidaceae*), and other genera including *Coptis chinensis*, *Hydrastis canadensis*, *Xanthorhiza simplicissima* (*Ranunculaceae*), *Phellodendron amurense* (*Rutaceae*), *Tinospora cordifolia* (*Menispermaceae*), *Argemone mexicana*, and *Eschscholzia californica* (*Papaveraceae*) [100]. It has a strong antimicrobial ability, broad antibacterial spectrum, and wide applicability prospect in the treatment of bacterial and fungal infections and other skin diseases [117]. The reduced solubility in both aqueous and lipidic environments makes berberine require other substances or methods to facilitate TD [118]. But, the importance of this alkaloid in the treatment of skin diseases, including psoriasis, has motivated a multitude of studies regarding appropriate trans epidermal transport methods and supports (nanocarrier, the impregnation of nanobiocellulose of a microbial origin, etc.) [119,120].

***Galantamine hydrobromide*** (***GH***) is an alkaloid that proves effective therapeutic management in memory impairment diseases, especially Alzheimer’s disease, but recent studies used this alkaloid as transdermal treatment for rheumatoid arthritis [121,122]. Galantamine is found in various plant sources, mainly from the genera *Amaryllis*, *Lycoris*, *Hippeastrum*, *Ungernia*, *Leucojum*, *Zephyranthes*, *Narcissus*, *Galanthus*, *Hymenocallis*, and *Haemanthus*, and is a naturally occurring alkaloid of the family Amaryllidaceae [123]. GH was originally isolated from plants, but now is chemically synthesized.

***Ephedrine*** is an alkaloid with a phenethylamine skeleton, obtained from the plant *Ephedra sinica* and other members of the genus *Ephedra*, from where the name of the substance derives [124]. Ephedrine causes cardiovascular effects similar to those of epinephrine: an increase of blood pressure, heart rate, and contractility. Like pseudoephedrine, it is a bronchodilator, but with a considerably higher effect. Ephedrine can reduce motion sickness, but it is mainly used to reduce the sedative effects of other motion sickness drugs combined in TP [125,126,127].

#### 4.2.2. Phenolic Compounds (PC)

***Phenolic compounds (PC)*** are SM widely distributed in the plant kingdom, found in various higher organs of different plants, such as medicinal plants, spices, vegetables, and cereals, with about 8000 different phenolic structures [128,129,130,131,132,133,134].

The most important role of these substances is that of plant adaptation in stressful conditions, such as damages, infections, or exposure to UV radiation, but they also may have secondary roles such as involving color, aroma, taste, etc. [128,132,135]. The presence of PC in the diet and in medical treatments is recognized in the prevention of many human diseases. Research carried out to date demonstrates the benefits of PC to the human body, such as antioxidant, antimicrobial, vasoactive, and anti-inflammatory effects, their ability to interact with enzymes and cellular receptors, etc. [136,137,138,139,140].

From a structural point of view, PC are substances that contain at least one benzene ring (C6) and one or more hydroxyl groups [128,135]. The metabolic pathways through which plants produce these SM are (i) the shikimate/chorismate or succinyl–benzoate pathway, which produces phenylpropanoid derivatives (C6-C3); (ii) the acetate/malonate or polyketide pathway, which produces side-chain elongated phenylpropanoids, including the large group of flavonoids (C6-C3-C6) and some quinones; and (iii) the acetate/mevalonate pathway, which produces aromatic terpenoids, mostly monoterpenes, through dehydrogenation reactions [128,135,141,142].

There are different criteria to classify PC, by the number of carbon atoms in the molecule, by the number of phenol units, or even according to the importance in the human diet [141,142]. Most classifications consider only the number of phenolic units in the structure of PC; however, some consider both the structure of PC and their pharmacological importance. In this review, the classification adopted by Nurzynska-Wierdak, 2023 is considered the most appropriate to describe the importance of PC in the dermal and TD of various human ailments. Thus, PC are divided into five subgroups [142]:-Phenolic acids (hydroxybenzoic and hydroxycinnamic acids).-Flavonoids (flavonols and flavan-3-ols, flavones, flavanones, isoflavones, flavanones and anthocyanidins).-Coumarins.-Lignans.-Stilbenes.

The bioavailability of PC is not very good; therefore, their inclusion in controlled drug release systems is a current research topic. Problems such as a poor solubility, stability, and permeability are to be overcome. The bioavailability of PC depends on the subclass of phenolic compounds and their physicochemical properties, such as the degree of polymerization or molecular properties, polarity, and interactions with different components of the formulation in which they are introduced [138,139,143].

##### Non-Flavonoid PC

***Phenolic acids*** (PA) (***hydroxybenzoic acids—HBA and hydroxycinnamic acids—HCA***)

HBAs have a C6-C1 main structure derived directly from benzoic acid and include p-HBAs such as proto-catechuic, vanillic, gallic, and syringic acids, while HCAs are aromatic compounds with a side chain of three carbon atoms (C6-C3), and they are represented by coumaric, caffeic, ferulic, and synaptic acids. PA are found in vegetables, and specific HBAs are abundantly found in oilseeds, cereals, coffee beans, cowpeas, blackcurrants, raspberries, skins and seeds of pumpkins, and blackberries, while HCAs are found in coffee beans, cherries, grains, peaches, spinach, citrus fruits, plums, tomatoes, potatoes, and almonds [142,144,145].

PA have both antioxidant and antimicrobial action, but there are not many studies on their individual action in TDD, specifically in TP. Most often, they are investigated together with other phenolic compounds, as natural plant extracts, in the transdermal treatment of wound healing as ion gels, nanoemulgels (anti-inflammatory treatments, skin disorders, diabetic gangrene), or microneedles for better TD [110,146,147,148,149,150,151,152].

***Curcuminoids*** are bright yellow pigments, representing a group of lipophilic diketones extracted from the rhizomes of *Curcuma longa*, a plant from the Zingiberaceae family, used for medicinal purposes for decades mostly in the Asian region. Curcumin, the main used curcuminoid, with antitumoral, antioxidant, antiarthritic, anti-amyloid, anti-ischemic, and anti-inflammatory properties, is poorly absorbed through skin barriers. Because of its hydrophobicity, low bioavailability, and chemical instability, different formulations are studied to improve its therapeutic efficacity including as nanoemulsions, nanohydrogels, nanoparticles in smart film and nanoparticles in hydrogels, nanostructured lipid carriers, lipid-based self-nanoemulsifying systems, ethosomes, and microneedles [90,153,154,155,156,157,158,159,160,161].

***Lignans*** are dimeric structures formed by a β-β′ covalent bond between two phenylpropanoid units. Neo-lignans are a subclass of lignans in which the covalent bond is replaced by any other type of connection, including oxygen etheric linking. All natural compounds that co-occur with lignans or neo-lignans, and possess C15, C16, or C17 core structures, are named nor-lignans [162]. Lignans show a large array of biological properties, including antioxidant, antifungal, antibacterial, and anti-inflammatory properties [163,164,165,166,167]. The most common biological source of lignans is flax (*Linum usitatissimum*) seeds [162,168]. However, these bioactive compounds are very lipophilic and their potential use in pharmacological studies is limited, because of a very low solubility in aqueous media [169,170]. The TD of lignans can be facilitated by sophorolipid-based transferosomal hydrogel or microemulgel in cosmetic application, or lignan-loaded solid lipid nanoparticles for topical application [170,171,172].

***Stilbenes*** belong to polyphenol group compounds and consist of two phenyl groups connected by ethene. They are found in fruit crops (especially grapes, peanuts, berries) and medicinal plants, including *Polygonum multiflorum*, *Polygonum cuspidatum* (Polygonaceae), *Hopea chinensis* (Dipterocarpaceae), *Gnetum parvifolium* (Gnetaceae), *Caragana sinica* (Leguminosae), *Morus alba* (Moraceae), etc. The most frequently used stilbene derivate compounds are pterostilbene (PTS) and resveratrol (RSV). Exploited now in medicine, stilbenes possess a variety of biological activities, including antitumoral, anti-inflammatory, antioxidant, antibacterial, and antiviral activities [11,165,173,174,175,176]. Several systems have been studied for the TD of stilbenes, such as polyvinyl pyrrolidone-based dissolving microneedle patches, or different types of nanocarriers: anionic phospholipids, polymeric nanoparticles, lipid base nanoparticles, cyclodextrin complexation, electrospun nanofibers, nanocrystals, liposomes, dendrimers, and niosomes [11,174,177,178,179,180].

##### Flavonoid PC 

**Flavonoid PC** are a class of substances composed of more than 6000 low-molecular-weight phenolic compounds with a skeleton of flavan. The main subgroups are flavones, flavonols, flavonones, flavononols, flavan-3-ols, anthocyanines, isoflavones, and chalcones. Flavonoids are substances with strong antioxidant, anti-inflammatory, and antiplatelet activities and are found in a very wide range of vegetables and fruits [181,182,183]. Among the many delivery systems developed so far, lipid-based nanoparticles, including liposomes and lipid nanoparticles, as well as polymer-based nanoparticles, are the most commonly used for the TD of flavonoids, because of their lipophilic nature and poor water solubility, which lead to a limited bioavailability.

The studies carried out on the use of flavonoids in TP focus in particular on the following subclasses:-*Flavanols* represented by a catechin group, containing epicatechin (EG) and specifically epigallocatechin gallate (EGCG), have attracted much attention due to their wide spectrum of biological properties, including antioxidant, photoprotective, antiviral, and antibacterial, as well as anticarcinogenic and neuroprotective, properties. EGCG is used as TP in the treatment of atopic dermatitis in the form of loaded microneedles, or with Ag nanoparticles in cotton patches for gangrene in diabetics [184,185]; EG is a TD using microneedles [114].-*Flavanones*, especially naringenin and hesperidin, are inserted into TP with an anti-inflammatory purpose. They are used for atopic dermatitis, WD nanoemulsions or nanogels on different polymeric supports, or lipidic nanocarriers [186,187,188].-*Flavones* represented by apigenin, which is a hydrophobic flavone, have been used in several formulations so far for TD, including liposomes, nanocrystal gel formulations, and self-micro-emulsifying delivery systems for anti-inflammatory and breast cancer treatment [189,190]. Luteolin is another promising flavonoid with antiarthritic activity. Topical formulations to treat psoriasis, or to keep breast cancer under control, were developed [191,192]. Due to its lipolytic character, the transport system is of a lipidic nanocarrier or niosome type [192,193,194].-*Flavonols*—quercetin is one of the most intensively studied and common flavonols found in nature; because this flavonoid shows a poor permeability in normal or excised human skin, it has been incorporated into various delivery systems, including nanoemulsions, nanocapsules, lipid nanoparticles, and microemulsions, to increase skin solubility and permeability [139,195]. TP with quercetin, incorporated in nanocomposite materials using microneedles, are used for the treatment of inflammation or androgenic alopecia [134,196]. Kaempferol is another well-known flavonol with antioxidant, anti-inflammatory, anticancer, and antiallergic properties, and it is used in the same TD systems compared to other flavonoids, mainly using lipid nanocarriers [197].-*Isoflavones*—represented mainly by genistein and daidzein, found in soybeans (*Glycine max*) and mung beans (*Vigna radiata*). Both are a TD with lipid nanocarriers [198,199].

For the TD of flavonoids, Zhang and Michniak-Kohn (2020) introduced the term “flavosomes”, the new deformable liposomes, which, in addition to the flavonoid compound, with the lipolytic structure, can be associated with an active synthetic API, to potentiate the anti-inflammatory action.

Additionally, a special class of flavonoids, intensively studied for TD in inflammation, cardiovascular diseases, or skin conditions, such as acne, is the licorice flavonoids, extracted from the root and rhizomes of the *Glycyrrhiza glabra* (Leguminosae family). These contain the same classes as the typical flavonoids, and their most important representatives include liquiritin, isoliquiritin, liquiritigenin, isoliquiritigenin, glabridin, licochalcone A, licoflavone A, licochalcone B, retrochalcone, and formononetin [194,200]. The TD of this type of flavonoids is achieved through microemulsions or hydrogels [194,201].

#### 4.2.3. Sulfur-Containing Compounds

***Sulfur-containing compounds*** can also be isolated from plants, and used for their diverse biological activities, such as anti-inflammatory, antioxidant, and anticancer activities [202,203,204,205,206,207,208].

Organosulfur natural products (ONP) refer to types of organic natural products that contain sulfur elements, such as thiols, thioesters, sulfoxides, etc., which play an important role in the pharmaceutical industry [209,210]. Although studies focus mainly on glucosinolates and S-alk(en)yl cysteine sulfoxides, there are other sulfur-containing secondary metabolites synthesized from plants that are of research interest (e.g., H_2_S, SO_2_, antimicrobial peptides, etc.) [211]. Glucosinolates are non-reactive sulfur-containing phytoanticipins found in plants of the family Brassicaceae (*Brassica oleracea* species, e.g., broccoli, cabbage, cauliflower, etc.), and S-alk(en)yl cysteine sulfoxides (ASCOs) are predominantly found in the Allium family of vegetables (i.e., *Allium sativum*, *Allium cepa*, *Allium porrum*), and their thiosulfinated derivatives are behind many of the flavor compounds of these vegetables [212]. Natural organic sulfur compounds are used in TDD combined with other pharmacological substances in lipid base nanocarriers for the treatment of wounds and psoriasis, as well as for antibacterial or antitumoral activity [213,214].

#### 4.2.4. Terpenoids 

***Terpenoids***—the largest class of plant SM represents about 60% of all known natural products. Over 40,000 different identified structures of terpenoids, or isoprenoids, are found in plants, and they are responsible for the scents, flavors, and colors [215]. In a TD system, essential oils (EO), included in this group, represent the most efficient systems for SC penetration and transdermal transport facilitation, being called “natural enhancers” [216,217,218,219,220].

This class of SM basically consists of five carbon isoprene units, assembled to each other (many isoprene units) in thousands of ways. Terpenes are simple hydrocarbons, while terpenoids are a modified class of terpenes, with isopentenyl pyrophosphate (IPP) as a structural unit, with different functional groups and an oxidized methyl group moved or removed at various positions [221].

Based on the number of isoprene units, terpenoids can be classified into hemiterpenoids (one isoprene unit—C5), and their occurrence is rare and they have no special biological significance; monoterpenoids (two isoprene units—C10); sesquiterpenoids (three isoprene units—C15); diterpenoids (four isoprene units—C20); sesterterpenoids (five isoprene units—C25); triterpenoids (six isoprene units—C30); tetraterpenoids (eight isoprene units—C40); and other terpenoid (5nC) isoprene unit condensation compounds [215,222]. As follows, the terpenoids used in WD and TDS are presented in brief.

-*Hemiterpenes* are the simplest terpenes, and the number of known compounds is lower than 100. Most hemiterpenes are in the form of oils and insoluble in water, with some glycosylated exceptions that are water soluble. The best-known hemiterpene is isoprene, which is the basic unit of all terpenes. Other known examples are tiglic, angelic acids, and isovaleric acid. These were used as enhancers for SC permeability in TP developed based on poly 3-hydroxybutyric acid-co-3-hydroxy valeric acid (PHBV) nanofibers, to deliver active substances (e.g., curcumin) for WD and to ensure an antimicrobial effect [223].-*Monoterpenes* are widely distributed in secretory tissues such as oil glands or resin chambers and ducts of higher plants. Monoterpenes are the major constituents of volatile plant oils. A number of monoterpenes are oxygenated and many exhibit biological and perfuming activities. Some are often used as medicinal agents or ingredients in cosmetic products. Monoterpenes frequently used as SC penetration agents are anethol, borneol, and campho (used in commercial TP for pain release); carvacol, carvone, and 1.8–1.4 cineol (used for TDDS in psychiatry); cymene, eugenol, fenchone, geraniol, limonene, linalool, and menthol (used as an enhancer and also for pain release); menthone, a-pinen oxide, pulegone, rose oxide, safranal, terpinene-4-ol, a-terpineol, tetra-hydrogeneraniol, and tymol (oral patches); and verbenone (perfumery and cosmetic use) [215,224,225,226,227,228,229,230,231,232,233,234,235,236,237,238,239,240,241,242].-*Sesquiterpenes* are the class of SM consisting of three isoprene units (C15) and are found in linear, cyclic, bicyclic, and tricyclic forms. Since sesquiterpenes have a high allergenic potential on the skin, their use in TP is limited [243,244,245].-*Diterpenoids* belong to a versatile class of chemical constituents found in different natural sources having four isoprene units. They can be classified as linear, bicyclic, tricyclic, tetracyclic, pentacyclic, or macrocyclic diterpenes, depending on their skeletal core. In nature, they are commonly found in a polyoxygenated form, with keto and hydroxyl groups, often esterified by small-sized aliphatic or aromatic acids [244]. This class of compounds show significant biological activities including anti-inflammatory, antimicrobial, anticancer, antifungal, cardiovascular, etc. [215]. Among tricyclic diterpenoids, abietic acid was used for wound healing and tanshinone IIA—with numerous therapeutic uses—was also included in TD systems as nanoparticles or microneedles [246,247,248].-*Triterpenoids* are a major class of SM and contain compounds with a carbon skeleton based on six isoprene units, derived biosynthetically from the acyclic C30 hydrocarbon, squalene. They have relatively complex cyclic structures, most being either alcohols, aldehydes, or carboxylic acids [215]. Sterols are triterpenes that are based on the cyclopentane perhydrophenanthrene ring system. Plant sterols called “phytosterols”, e.g., sitosterol, stigmasterol, and campesterol, are widespread in higher plants. Triterpenes have many active sites for glycosylation, which converts them into another class of compounds, namely, saponins (triterpene glycoside). Examples of triterpenoids of medical interest are squalene used in scaffold emulgel for wound healing, ursolic acid used in niosomal gel systems for arthritis treatment and skin diseases, oleanolic acid used in nanocarriers in TD, and sitosterol used in alopecia treatment with microneedle enhancers, or in breast cancer as nanoparticles [249,250,251,252,253,254,255].-*Tetraterpenoids* consist of eight isoprene units and 25 carbon atoms. The most common tetraterpenoids are carotenoids, which are natural fat soluble pigments [244]. Carotenoids are extremely hydrophobic molecules, with little or no solubility in water, and a strong tendency to aggregate in aqueous solutions. Since carotenoids are unstable at high temperatures, in the presence of light and oxygen, their incorporation into micro- and nanostructures is necessary, to increase their stability and preserve their bioactivity [256,257]. Examples of used carotenoids are as follows: lycopene loaded in a lipid nanocarrier in TDD systems; β-carotene in TDS as self-nanoemulsion or as lipid base nanocarriers, or as hydrogel with plant extracts; lutein with a high antioxidant effect is delivered in TDS using nanostructured lipid carriers, solid lipid nanoparticles, a nanosphere, a liposome, nanoemulsions, and polymeric nanoparticles for eye diseases, for anti-inflammatory effects, obesity, etc.; and crocin in a nanocellulose membrane of microbial cellulose-based TP [258,259,260,261,262,263,264].-*Polyterpenoids* are polymeric isoprenoid hydrocarbons, which consist of more than eight isoprene units. This class of compounds include rubber and gutta percha, used as polymeric matrices for TDDS [16,265].

Table 2, presented as follows, highlights the most recent medical applications (only 2023 publication year) of SM derived from plants, in topical or TD treatments, for various ailments or wound treatment.

## 5. Conclusions

Currently, medical devices for transdermal delivery and wound dressing materials are of a wide diversity, especially regarding the way of the transport and delivery of the active pharmaceutical ingredients to the skin and through the *Stratum Corneum*. Plants’ primary and secondary metabolites are solely used or combined for a favorable medical diagnosis, both at the level of the skin and endogenously targeted at certain types of tissues.

This review shows how plants were broken down into active principles or nonactive substances, and according to the different types of these compounds’ classes, how they were used for the treatment of various ailments. In ancient times, a plant was used as such (fresh or dried), or infused and impregnated on various textile supports, thus contributing with its entire arsenal of primary and secondary metabolites, as well as related microbiota, to the treatment of various diseases. Presently, technological possibilities allow the advanced extraction and purification of active compounds from plants, which contribute to a more efficient assembly of medical devices, better transdermal transport yields, and well-established orientation towards the target.

As new separation and preparation technologies can now offer a high purity and nanosized drugs, it is certain that new delivery vectors, and consequently effective treatment systems, should and can be developed in the near future. In a constantly increasing market of skin patches and transdermal drug delivery devices, interdisciplinary efforts are further required to develop strategies for the easier and safer administration of natural bioactive compounds.

## Figures and Tables

**Figure 1 plants-12-02661-f001:**
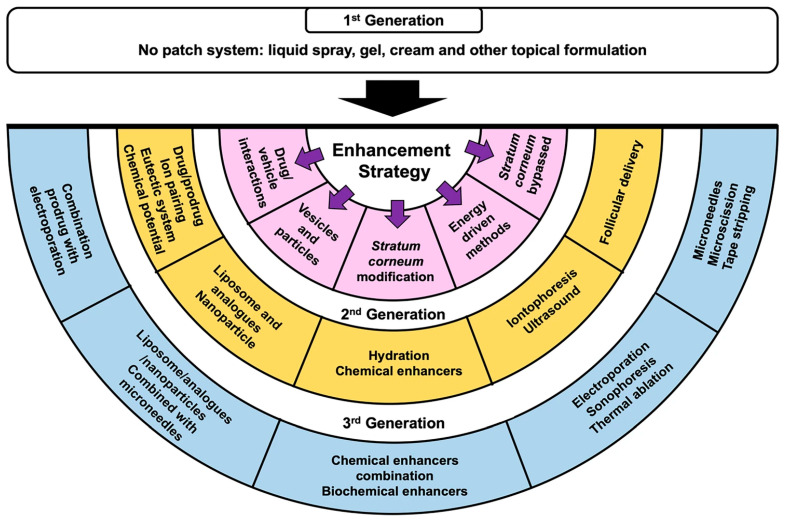
Five main methods to enhance API absorption across the skin: drug–vehicle interaction, vesicles, and analogues; SC modification; energy-driven methods; and SC bypass [16]. (Pink color indicates the first generation of SC penetration devices; Yellow—the second generation of SC penetration devices/methods; Blue—the third generation of SC penetration devices/methods).

**Figure 2 plants-12-02661-f002:**
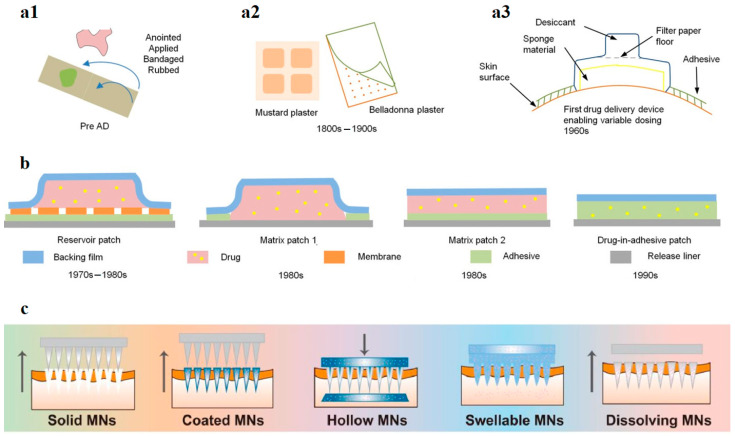
Evolution of patches: (**a1**) ointment on a bandage system representation from ancient times; (**a2**) plaster (here, mustard and belladonna for controlled drug delivery, medical device developed prior to 1900s); (**a3**) delivery device (here, first recognized topical delivery system); (**b**) different types of patches; (**c**) different types of microneedles (minimally invasive patches). Adapted with permission [22,23]. (The arrows represent the application/removal method of the microneedle device).

**Table 1 plants-12-02661-t001:** Mucilage-based products for WD and API delivery applications.

Mucilage Plant Source	Other Constituents	Active Ingredient/Drug	Applicability and Properties	Ref.
*Ficus reticuleta* fruit	Glycerol Propylene Glycol Span-80 Propyl/Methyl paraben	Diltiazem HCl	Therapy of hypertension, angina pectoris, arrhythmia Controlled release matrix	[40]
*Ficus carica* fruit	Propylene glycol Span-80 Propyl/Methyl paraben	Pioglitazone HCl	Type II diabetes therapy Matrix-moderated transdermal patch Retardation of release	[41]
	Povidone Propylene glycol Span-80 Propyl/Methyl paraben	Tramadol HCl	Patch for therapy of pain, inflammation, and arthritis Sustained drug release	[42]
*Artocarpus heterophyllus* fruit	Gelatin Glycerol Dimethyl sulphoxide	Naproxen sodium	Therapy for rheumatoid arthritis, osteoarthritis, and ankylosis spondylitis Sustained drug release	[43]
*Ficus benghalensis* fruit	Eudragit L Propylene glycol Span-60	Aceclofenac	Treatment of pain and inflammation Controlled discharge matrix transdermal patches	[44]
*Ficus auriculata* fruit	Hydroxypropyl Methyl Cellulose K4M Polyethylene glycol-400 Tween-80	Diclofenac Potassium	Treatment of arthritis symptoms Matrix-moderated transdermal patch Controlled release diffusion	[45]
*Ocimum basilicum* seeds	Polyvinyl alcohol Glutaraldehyde Glycerol	Tetracycline Hydrochloride	Wound dressing in chronic wound management pH-sensitive cross-linked hydrogel Drug release control	[46]
	Borax Glycerol	Zinc oxide nanoparticles	Wound dressing Crosslinked antimicrobial sponge	[47]
	L-cysteine-modified alginate Dopamine-modified hyaluronic acid Thiolated alginate Polydopamine	Nystatin	Oral fungal infection treatment Bi- and tri-layer mucoadhesive sublingual films Controlled release	[48]
*Cydonia oblonga* seeds	Chitosan Poly-caprolactone Polyethylene glycol		Skin-tissue-engineered hybrid scaffolds Dressing for continuous absorption of wound exudates Smart/stimuli-responsive bio-matrix	[49]
	Bacterial cellulose		Composite scaffold for wound dressings	[50]
Leaves of *Cocculus hirsutus*	Polyvinyl alcohol		Dermal wound patches Biodegradable biopolymer	[51]
*Salvia hispanica* seeds			Dermatological and cosmetic formulations Biopolymer with mucoadhesive properties Controlled release of bioactive compounds and drugs	[52]
*Lallemantia royleana* seeds	Chitosan Chitin	Ag nanoparticles Ciprofloxacin	Wound dressings Controlled drug release	[53]
*Althea rosea* plant			Functional and nutraceutical in inflammatory disorders	[54]
*Aloe vera* leaves (fresh or powdered)	Sodium alginate Glycerol		Improved film thermal stability and transparency Good UV light barrier Suitable for wound healing and drug delivery	[55]
	Hydroxypropyl methylcellulose PEG 400	Captopril	Improved drug transdermal penetration	[56]
	Xanthan gum	Allantoin Natural BHA salicylic acid	Anti-inflammatory, wound epithelization, and skin regeneration hydrogel	[57]

**Table 2 plants-12-02661-t002:** Plant-SM-based products for WD and API delivery applications.

SM Plant Source	Main SM	TDS	Applicability and Properties	Ref.
*Mitragyna Speciosa* leaves’ extract	Alkaloids Flavonoids Saponins	Lipidic carriers in oleic acid, Tween 80	For topical application/wound dressing	[266]
*Pistacia atlantica* fruit extract	β–sitosterol	Lipidic carriers: span 40, pluronic F127, Cholesterol and Glycerol	Antimicrobial in topical application	[267]
*Eysenhardtia platycarpa*	Flavone	Biopolymeric nanoparticles	Topical application for cancer treatment	[268]
*Lippia origanoides, Turnera diffusa*	Eugenol Carvacrol Limonene	Carbopol hydrogel	Caffeine’s skin permeation for inflammatory diseases	[227]
*Catharanthus roseus*	Vinblastine Vincristine (alkaloids)	Lipid nanocarrier	Topical treatment of breast cancer	[269]
*Mucuna prurita*	L-Dopa			
*Rauvolfia serpentine*	Reserpine			
*Taxus brevifolia*	Paclitaxel			
*Vigna radiata*	Flavonoids Phenolic acids	Aloe vera hydrogel	Topical treatment of acne vulgaris	[270]
*Curcuma longa*	Curcumin	Loaded in biopolymers (CMC-silk sericin)	Antibacterial, antioxidant, and anti-inflammatory for wound healing	[271]
		Nanoparticles	Different types of cancer	[272]
		Chitosan sponge—cryogel	Wound healing	[273]
		Nanofibrous meshes	Wound healing	[274]
*Cannabis sativa*	Cannabinoids Terpenoids Sterols Flavonoids	Nano emulsions (Tween 80, PEG 400)	Transmucosal delivery	[275]
*Centella asiatica*	Flavonoids Plant sterols Eugenol Pentacyclic triterpenoids	Hydroalcoholic extracts	For topical application/WD	[276]
		Nanofibrous meshes		[274]
*Chrysanthemum indicum var. aromaticum*	Terpenoids Flavonoids Phenols	Gel formulated with Tween-80, transcutol-P, sodium hyaluronate, and glycerol	Topical application for inflammatory skin disease, cosmetic use	[277]
*Zingiber officinale*	Sesquiterpene Monoterpene hydrocarbons	Nanocarriers	Major inflammatory disease	[278]
*Sideroxylon mascatense*	Berberine Palmatine Jatrorrhizine Saponins	Nanoemulsion gel	Wound healing	[279]
*Jasione montana*	Luteolin Luteolin 7-O-glucoside Luteolin 7-O-sambubioside	Direct application (scratch cells)	Wound healing	[280]
*Caesalpinia coriaria*	*Corilagin* (ellagitannin)	Gelatin/agar microsphere	Wound healing	[281]
Genus *Sophora*	Oxymatrine natural alkaloid	Deep eutectic solvent for TD Capric acid, Lauric acid, Myristic acid, Palmitic acid, and Stearic acid	Anti-pruritic and anti-inflammatory effects	[282]
*Calendula officinalis*	Carotenoids Lycopene Phenolic acids Hydroxycinnamic acids Flavonoids (rutin) Coumarins (esculetin)	Polyacrylamide hydrogel containing calendula extract	Therapeutic applications, which include tissue re-epithelialization and general wound healing action	[283]
*Carissa carandas* leaf extract	Ursolic acid Phenolic and flavonoid compounds	Liquid crystals composed of 2.5% *w*/*w* coconut oil, 30% *w*/*w* Tween^®^ 85	Antioxidant, anti-inflammatory, and antiaging effects, decrease inflammatory cytokine and matrix metalloproteinase (MMP) production, as well as enhance the level of ceramide and collagen in the skin	[284]
*Epilobium angustifolium*	Polyphenols (flavonoids, phenolic acids, and tannins)	Biocellulose	Anticancer properties	[285]
*Stereocaulon japonicum*	Atraric acid Phenolic compound	Volatile and nonvolatile solvents, as well as oleic acid, an effective permeation enhancer	Androgenic alopecia	[286]
*Azadirachta indica*	Alkaloids	Nanofibrous meshes	Wound healing	[274]
*Tecomella undulata*	Alkaloids	Nanofibrous meshes	Wound healing	
*Hypericum perforatum,*	Polyprenylated phloroglucinol hyperforin	Bigel nanocarriers	Wound healing	[287]
*Dodonaea viscosa*, leaf extract	Flavonoids (quercetin and kaempferol)	Direct application (scratch cells)	Wound healing	[288]

## Data Availability

No new data were created or analyzed in this study. Data sharing is not applicable to this article.

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
