# Peer review of "From Plants to Wound Dressing and Transdermal Delivery of Bioactive Compounds"

_plants, 2023, doi:10.3390/plants12142661_

Round 1

Reviewer 1 Report

Plants-2429193 "From plants to bioactive transdermal patches" describes the application of plants source were constructed to satisfy skin patches. Although the study is interesting, there are still some areas to improve before it can be published in the Plants:

1. The abstract is not comprehensive enough and logic is confusing, which should be rewrited.

2. The title showed be rewrited, which can’t reflect the full text.

3. The content of the manuscripts in logic is confusion.

No

Reviewer 2 Report

The manuscript is disorganized and it is hard to follow the flow. The abstract is very general and there is no specific highlights on bioactive transdermal patches. Section 1 and 2 are textbook knowledge and the figures mainly derived from other studies. Moreover, In Table 1, some of them are not related to transdermal drug delivery.  Section 3 is very lengthy and it is not known how these secondary metabolites contribute to TDD.

Moderate editing of English is required.

Reviewer 3 Report

Dear Authors,

This is a well written article, however, there are a few things that can be improved to increase general readability.

·         In abstract, the plants & traditional medicine introductions are quite long (they are not even mentioned in text of the article). Instead, TDS products needs more explanation.

·         Minor typos in the text

·         Authors choose AC as an abbreviation for active. The standard abbreviation is API – Active pharmaceutical ingredient. Please replace in the entire document.

·         Recent literature on skin and skin products including microneedles can be added to improve the introduction section. Suggestions -https://doi.org/10.1016/j.addr.2022.114293

·         Figure 1 and 2 can be redrawn. Especially, figure 2 which is wrongly associated with a 1961 publication. Please check.

·         Interesting points about plants mucilage.

·         Outlines and section segregation are clear and easy to follow

·         Figures can be described in more detail in the legends

Style of writing can be improved. Especially in the abstract, which is too concise and in adequate. 
